# Accelerated PDEs for Construction and Theoretical Analysis of an SGD Extension

**Yuxin Sun**[1]    **Dong Lao**[2]    **Ganesh Sundaramoorthi**[3]    **Anthony Yezzi**[1]

[1]Georgia Institute of Technology,    [2]UCLA & KAUST,    [3]Raytheon Technologies

{syuxin3,ayezzi}@gatech.edu, lao@cs.ucla.edu, ganesh.sundaramoorthi@rtx.com

## Abstract

We introduce a recently developed framework (*PDE acceleration*), which is a variational approach to accelerated optimization with partial differential equations (PDE), in the context of optimization of deep networks. We derive the PDE evolution equations for optimization of general loss functions using this variational approach. We propose discretizations of these PDE based on numerical PDE discretizations, and establish a mapping between these discretizations and stochastic gradient descent (SGD). We show that our framework can give rise to new PDEs that can be mapped to new optimization algorithms, and thus theoretical insights from the PDE domain can be used to analyze optimization algorithms. We show an example by introducing a new PDE with diffusion that naturally arises from the viscosity solution, which translates to a novel extension of SGD. We analytically analyze the stability and convergence using Von-Neumann analysis. We apply the proposed extension to optimization of convolutional neural networks (CNNs). We empirically validate the theory and evaluate our new extension on image classification showing empirical improvement over SGD.

## 1   Introduction

Acceleration (momentum) and Nesterov acceleration are now standard in optimizing deep networks by stochastic gradient descent (SGD) and its variants, although the underlying principles of these methods are still under development. Understanding the principles behind these methods could lead to new improved optimization methods. To this end, [27] showed that Nesterov's accelerated optimization [15] can be formulated as a discretization of an ordinary differential equation (ODE) [23] arising from a variational principle, i.e., the Principle of Least Action in classical physics. Several new optimization schemes have since arisen from different discretizations of the same ODE [22]. An extension of this approach from the ODE framework into the PDE framework was done in [24, 3], which allowed for the accelerated optimization of multi-dimensional problems. New optimization methods were formulated arising in particular from various discretizations of the underlying PDE [3].

In this work, we study SGD with momentum within the framework of Accelerated PDE and PDE numerical discretization. We establish a mapping from SGD with momentum to Accelerated PDE. This allows one to derive new extensions of SGD by translating new PDE naturally arising within the Accelerated PDE framework into numerical discretizations, which map back to extensions of SGD. We propose a novel extension of SGD in this way, and apply it to optimization of CNNs. The derivation of the method from numerical discretization of Accelerated PDE allows us to establish stability and convergence results. More specifically, our contributions are

1. We show how SGD with momentum can be related to discretizations of an Accelerated PDE.

2. Within this Accelerated PDE framework, we introduce a diffusion term arising from the viscosity solution (2) of the PDE.

35th Conference on Neural Information Processing Systems (NeurIPS 2021), Sydney, Australia.

3. We show how the discretization of the viscosity solution of this Accelerated PDE translates to a simple modification of the usual SGD with momentum, leading to a new optimization scheme.

4. We exploit the discretizations of the Accelerated PDE to show that our new optimization scheme is stable/convergent provided that the original SGD with momentum scheme is stable/convergent.

## 1.1 Related Work

Many extensions of SGD (e.g. [7, 29, 10, 10, 2, 13]) have been proposed (see [4] for a survey) and there has been rigorous analysis of some of these methods (e.g. [21]). A framework to derive methods could lead to further understanding and new optimization schemes. Our goal is to take a small step in this direction by applying the Accelerated PDE framework [24] to study SGD with momentum under discretizations of PDE. In this regard, Wibisono, Wilson, and Jordan [27] showed that all Nesterov type accelerated descent methods [15, 18, 19, 16, 17] can be realized as discretizations of equations of motion (ODE) arising from an Action Integral of a generalized Lagrangian. Following the Lagrangian formulation, [24, 3] developed an accelerated PDE framework and applied it to computer vision problems. [24] introduced the use of diffusion motivated from Viscosity Theory [5], allowing the use of entropy discretizations to address shocks/fans in the resulting PDE. The PDE acceleration framework was further developed in [3] in the context of calculus of variations problems defined for functions on $\mathbb{R}^n$, including stability analysis for various discretization schemes. We apply discretization methods considered there to the novel accelerated PDE that we consider in this paper.

In application to CNNs, our method induces smoothness into optimization by diffusing the velocity, which has an effect similar to gradient smoothing. [20] considers smoothing the stochastic gradient by applying a Laplacian smoothing operator. [11] smooths along certain 1D directions of convolutional weight tensors using Sobolev gradients. While both methods [20, 11] are 1D, our diffusion can induce multi-dimensional smoothing with little implementation effort and computational cost.

## 2 Continuum PDE Acceleration Theory

We consider minimizing a loss function of the form $U : \mathcal{U} \to \mathbb{R}$ where $\mathcal{U} = \{u : \Omega \subset \mathbb{R}^N \to \mathbb{R}\}$ is a set of $N$-dimensional functions. In subsequent sections, $\mathcal{U}$ will represent a continuum version of the space of weight tensors at a particular layer of a network. The argument of $u$, denoted $x$, will be called the "spatial" argument. We derive the continuum evolution equations for accelerated optimization, which are PDE, and discretize the PDE in the next section for numerical implementation.

The formulation below to derive the optimization equations of a loss function is derived from Hamilton's principle of least action in classical physics [1, 14]. Following [24], we start with an action integral, defined on possible optimization paths $u_t : \Omega \to \mathbb{R}$ of $U$, where $t$ denotes the parameter (time) along the path, and its velocity (in the tangent space of $\mathcal{U}$) $V_t : \Omega \to \mathbb{R}$. We suppress $t$ from now on for simplicity of notation. The action integral is defined as

$$A = \int [\tilde{a}T(V) - \tilde{b}U(u)]dt, \tag{1}$$

where the path and its velocity are related as $\partial_t u = V$, $\tilde{a}, \tilde{b}$ are functions only of time that give rise to dissipative forces and will be specified, $T(V)$ is the kinetic energy, and $U$ is the loss function to be minimized, analogous to the potential. For the purpose of this paper, the kinetic energy is defined as $T(V) = \int_\Omega \frac{1}{2}\rho(x)|V(x)|^2 dx$, where $\rho(x)$ is the mass density, which for this paper is chosen to be 1.

The stationary conditions are derived by computing the first variation of $A$ and result in the equations:

$$\partial_t u = V, \quad \partial_t V = -\nabla U(u) - aV + b\nabla^2 V, \tag{2}$$

where $\partial_t$ denotes the time derivative, $\nabla$ is the spatial gradient, $\nabla^2$ is the spatial Laplacian, $a = \partial_t \tilde{a}/\tilde{a}$ is chosen to be a positive constant, $\tilde{b}$ is chosen to be 1, and $b > 0$. Note that the diffusion term (Laplacian term) results from the PDE viscosity solution to ensure existence of a solution [5]. The above system of equations is a forward (in time) evolution that starts with initial conditions on $u$ and $V$. We may write the above first order PDE system as a second order in time PDE, which results in

$$\partial_{tt}u + a\partial_t u = -\nabla U(u) + b\nabla^2 \partial_t u, \tag{3}$$

where $a, b > 0$ are considered hyperparameters of the optimization, with $a$ interpreted as a damping coefficient that leads to convergence, and $b$ interpreted as a diffusive coefficient. The above equation

resembles a wave equation, except for the fact that the spatial Laplacian term is on the velocity rather than $u$; this induces regularity on the velocity or the optimization in contrast to regularity within the loss function, which would have led to the standard wave equation.

# 3 Discretization of Accelerated PDE, Relation to SGD, and Analysis

## 3.1 Semi-Implicit Euler Discretization of the Accelerated PDE

We discretize the first two time derivatives in (3) using central differences, and the last time derivative with a backward difference. The last time derivative uses a backward difference as a central difference with the spatial derivative approximations would require an implicit scheme, requiring solution of a linear system, which we would like to avoid for simplicity.

To obtain systems which more closely resemble the classic two-part Nesterov recursion, we propose a semi-implicit Euler style discretization as considered in [3]. To do this, we replace the explicit discretization $\nabla U^n$ of the gradient with a "predicted estimate", $\widehat{\nabla U^{n+1}}$, of $\nabla U^{n+1}$ evaluated at a projected location of $u^n$, denoted $w^n$, as follows:

$$\begin{cases} w^n = u^n + \frac{2-a\Delta t}{2+a\Delta t}\Delta u^{n-1} + \frac{2\Delta t}{2+a\Delta t}b\nabla^2\Delta u^{n-1} \\ u^{n+1} = w^n - \frac{2\Delta t^2}{2+a\Delta t}\nabla U\left(w^n\right) \end{cases} \tag{4}$$

where $\Delta u^n = u^n - u^{n-1}$, and $\Delta t$ is the time-step of the time discretizations. The spatial Laplacian above is discretized using its standard discretization as

$$\nabla^2 d(x) = \sum_{k=1}^{N} \frac{d(x + e_k\Delta x) - 2d(x) + d(x - e_k\Delta x)}{\Delta x^2}, \quad d = \Delta u^{n-1} \tag{5}$$

where $x$ is the (spatial) argument of the tensor, $e_k$ is the unit vector in the $k^{\text{th}}$ direction, and $\Delta x$ is the spatial discretization amount (assumed to be uniform in all $N$ directions).

## 3.2 Relation to SGD With Nesterov Momentum

We establish a link between the Semi-Implicit Euler discretization (4) of the Accelerated PDE and SGD, which shows how quantities in the PDE discretization (e.g., step size, damping) translate to the quantities in SGD (e.g., learning rate, momentum), and vice-versa. This shows how the discretization of the Accelerated PDE (3), which gives rise to a novel extension of SGD, can be implemented with conventional deep learning frameworks.

**Theorem 1.** *The semi-implicit scheme* (4) *can be written as the following modification of SGD with Nesterov momentum[1] (SGD-Nesterov):*

$$v^{n+1} = \gamma v^n + \nabla_\theta J(\theta^n) + \beta\nabla^2 v^n \tag{6}$$

$$\theta^{n+1} = \theta^n - \eta[\nabla J(\theta^n) + \gamma v^{n+1} + \beta\nabla^2 v^{n+1}] \tag{7}$$

*where $\gamma, \eta, \beta$ are functions of the time step $\Delta t$ and damping $a$. $\gamma$ is the momentum, and $\eta$ is the learning rate. The mapping of SGD-Nesterov to our Semi-Implicit Euler Scheme* (4) *is done by setting*

$$\theta^n = w^n, \quad v^{n+1} = -\frac{2+a\Delta t}{2\Delta t^2}\Delta u^n, \quad \eta = \frac{2\Delta t^2}{2+a\Delta t}, \quad \gamma = \frac{2-a\Delta t}{2+a\Delta t}, \quad \beta = \frac{2b\Delta t}{2+a\Delta t}, \tag{8}$$

*and $J = U$. Notice in the case that the diffusion coefficient is set to $b = 0$ ($\beta = 0$), (6), (7) reduce to standard SGD with Nesterov momentum.*

The mapping from SGD-Nesterov to the Semi-Implicit Euler Scheme (4) as

$$w^n = \theta^n, \quad \Delta u^n = -\eta v^{n+1}, \quad \Delta t = \sqrt{\frac{2\cdot\eta}{\gamma+1}}, \quad a = \frac{2-2\cdot\gamma}{\sqrt{2\cdot\eta\cdot(\gamma+1)}}. \tag{9}$$

---

[1]Note the form above is Pytorch's version of Nesterov momentum, which is equivalent to the classical version in [25], see Appendix E.

### 3.3 Stability and Convergence Analysis

Using the mapping between the Accelerated PDE and SGD, we are able to use methods of analysis in numerical PDE to analyze properties of our new SGD extension, in particular stability and convergence of the new SGD extension. We compute the Courant–Friedrichs–Lewy (CFL) condition of the schemes, which are necessary conditions for convergence, to analyze the stability and convergence of the discretizations proposed in the previous section by applying Von-Neuman analysis [26], which involves analysis in the Fourier domain. By doing so, we can show the following theorem.

**Theorem 2** (CFL Conditions for Accelerated Discretizations)**.** *The Semi-Implicit Euler* (4) *Scheme with addition of diffusion term will maintain stability and converge provided that the time step $\Delta t$, the spatial step $\Delta x$, and the diffusion coefficient $b$ satisfy the relation:*

$$b\Delta t \le \frac{1}{2N}\Delta x^2, \tag{10}$$

*where $N$ is the dimension of the functions in $\mathcal{U}$.*

The maximum diffusion $b$ ($\beta$) for a given time step that yields a stable/convergent scheme is

$$b_{max} = \frac{1}{2N\Delta t} \implies \beta_{max} = \frac{1+\gamma}{4N} \tag{11}$$

where we have chosen $\Delta x = 1$. We experiment with varying amounts of diffusion relative to $\beta_{max}$.

## 4 Experiments

### 4.1 Empirical Validation of Stability Analysis

We empirically validate the condition in Theorem 2. We use the Cifar10 dataset and ResNet56 [8] with diffusion added to the BN weight tensor velocity (other tensors and architectures show similar behavior) and with the rest of the network trained on standard SGD with Nesterov. As shown in Figure 1, the training is stable with the maximum stable diffusion, $\beta_{max}$. However, if we raise the diffusion amount to over $\beta_{max}$, the training becomes unstable and fails to converge. The experiment shows the accuracy of the bound in Theorem 2 and the validity of PDE analysis for analyzing a deep learning optimization algorithm.

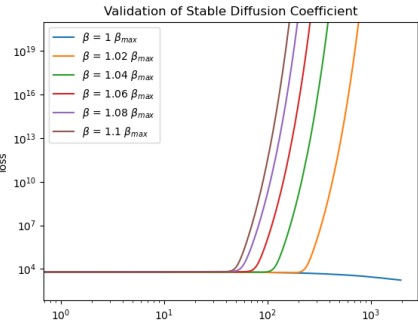

Figure 1: Stability of SGD extension

### 4.2 Empirical Analysis of Diffusion Effect

We analyze the behavior of the diffusion term on various choices of velocity tensors in optimization of a CNN. We study diffusion over various dimensions of 4D conv tensors, i.e., spatial dimensions of kernels, input/output directions, batch norm tensors. To isolate the effect of diffusion from empirically chosen hyperparameters (e.g., learning rate schedules) that are optimized for standard SGD, we study the case of fixed learning rates, which is also more common in PDE-driven optimization methods.

Test accuracy (average over 10 trials) against SGD-Nesterov (no diffusion) for Cifar-10 on ResNet-56 are shown in Table 1 (diffusion amounts are in terms of $\beta_{max}$ and amounts above $\beta_{max}$ are achieved by lowering the step size) and Table 2. For diffusion coefficients which exceed the maximum stable diffusion for a specific time step, the time is reduced to ensure the stability of the system. For 2 and 4 times of the maximum stable diffusion, we divide the time step by 2 and 4 respectively and recalculate the corresponding learning rate and momentum with equation(4). Other hyperparameters (e.g., momentum) are set to current practice. All diffusion amounts improve performance for batch norm layers over all learning rates and conv directions except at small learning rates. We are currently studying how to exploit the larger boosts for large learning rates to construct schedules for the learning rate and diffusion. In the meantime, with current schedules and training techniques optimized for standard optimizers, we do see statistically significant improved performance for diffusion to batch norm layers (see appendix), though at a diminished level.

| Layer | Diffusion | | | | | | Layer | Learning Rate | | |
|---|---|---|---|---|---|---|---|---|---|---|
| | 0.25 | 0.5 | 1 | 2 | 4 | | | 0.2 | 0.1 | 0.01 |
| None(SGD) | 80.52 | 80.52 | 80.52 | 80.52 | 80.52 | | None(SGD) | 80.52 | 85.84 | 90.25 |
| Conv Output | 84.90 | 85.62 | 86.15 | 87.02 | 87.70 | | Conv Output | 86.15 | 88.91 | 86.87 |
| Conv Input | 83.72 | 83.45 | 83.00 | 84.20 | 84.55 | | Conv Input | 83.00 | 86.63 | 85.64 |
| Conv Spatial | 81.50 | 82.97 | 81.50 | 81.66 | 81.29 | | Conv Spatial | 81.50 | 86.04 | 88.42 |
| BN Weight | 81.54 | 82.28 | 81.51 | 82.61 | 83.78 | | BN Weight | 81.51 | 86.49 | 90.27 |
| BN Bias | 81.04 | 81.43 | 80.81 | 81.57 | 82.01 | | BN Bias | 80.81 | 86.11 | 90.32 |

Table 1: Fixed learning rate of 0.2 results.      Table 2: Fixed diffusion of $\beta_{max}$ results.

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

## A    Proof for theorem 1

*Proof.* First we rewrite the semi-implicit scheme (4) in a Nesterov form. Replacing $u^n$ in the first equation with second equation we obtain:

$$w^{n+1} = w^n - \frac{2\Delta t^2}{2 + a\Delta t}\nabla U\left(w^n\right) + \frac{2 - a\Delta t}{2 + a\Delta t}\Delta u^n + \frac{2\Delta t}{2 + a\Delta t}b\nabla^2\Delta u^n \tag{12}$$

Substituting $w^n$ in the second equation into the first equation, we obtain:

$$\Delta u^n = \frac{2 - a\Delta t}{2 + a\Delta t}\Delta u^{n-1} - \frac{2\Delta t^2}{2 + a\Delta t}\nabla U\left(w^n\right) + \frac{2\Delta t}{2 + a\Delta t}b\nabla^2\Delta u^{n-1} \tag{13}$$

We now start to map the semi-implicit scheme to SGD with Nesterov momentum using the standard form in PyTorch. This is in following form:

$$\begin{cases} v^{n+1} & = \gamma v^n + \nabla_\theta J\left(\theta^n\right) \\ \theta^{n+1} & = \theta^n - \eta\left(\nabla_\theta J\left(\theta^n\right) + \gamma v^{n+1}\right) \end{cases} \tag{14}$$

where $\theta$ is the parameter and $v$ is the momentum buffer, $\eta$ for learning rate and $\gamma$ for momentum.

Compared to the equation (12) and (13), we formulate the Nesterov momentum with diffusion as

$$\begin{cases} v^{n+1} & = \frac{2-a\Delta t}{2+a\Delta t}v^n + \nabla_\theta J\left(\theta^n\right) + \frac{2\Delta t}{2+a\Delta t}b\nabla^2 v^n \\ \theta^{n+1} & = \theta^n - \frac{2\Delta t^2}{2+a\Delta t}\left(\nabla_\theta J\left(\theta^n\right) + \frac{2-a\Delta t}{2+a\Delta t}v^{n+1} + \frac{2\Delta t}{2+a\Delta t}b\nabla^2 v^{n+1}\right) \end{cases} \tag{15}$$

Note we can also obtain the relationship between the variables in the two different schemes:

$$\begin{cases} \theta^n & = w^n \\ v^n & = -\frac{2+a\Delta t}{2\Delta t^2}\Delta u^{n-1} \\ \eta & = \frac{2\Delta t^2}{2+a\Delta t} \\ \gamma & = \frac{2-a\Delta t}{2+a\Delta t} \end{cases} \quad \begin{cases} w^n & = \theta^n \\ \Delta u^n & = -\eta v^{n+1} \\ \Delta t & = \sqrt{\frac{2\eta}{\gamma+1}} \\ a & = \frac{2-2\gamma}{\sqrt{2\eta(\gamma+1)}} \end{cases} \tag{16}$$

$\square$

# B    Proof for theorem 2

*Proof.* We base our analysis on the homogeneous part of the PDE, and neglect the inhomegenous part of the PDE, i.e., the gradient component $\nabla U$, by setting it to zero [2]. In this way, the PDE becomes linear and we may apply the Discrete Fourier Transform (DFT) to the update schemes, which in the absence of the gradient are identical.

Taking the DFT with respect to $x$ on both sides of the homogeneous part yields

$$
\begin{aligned}
(2 + a\Delta t)\hat{u}^{n+1}(\omega) = \quad & [4 + 4\tfrac{b\Delta t}{\Delta x^2}\sum_{k=1}^{N}(\cos\omega_k\Delta x - 1)]\hat{u}^n(\omega) - \\
& [2 - a\Delta t + 4\tfrac{b\Delta t}{\Delta x^2}\sum_{k=1}^{N}(\cos\omega_k\Delta x - 1)]\hat{u}^{n-1}(\omega)
\end{aligned}, \tag{17}
$$

where $\omega = (\omega_1, \ldots, \omega_N)$ represents the spatial frequency, and $\hat{u}^n$ represents the DFT of $u^n$. Assuming a complex amplifier $\xi(\omega)$, we may substitute $\hat{u}^{n+1}(\omega) = \xi(\omega)\hat{u}^n(\omega)$ for all $n$, and thus find the following equation for $\xi$:

$$
\overbrace{(2 + a\Delta t)}^{A}\xi^2(\omega) - \overbrace{[4 + 4g(\omega)]}^{B}\xi(\omega) + \underbrace{[2 - a\Delta t + 4g(\omega)]}_{C} = 0, \;\; g(\omega) = \frac{b\Delta t}{\Delta x^2}\sum_{k=1}^{N}(\cos\omega_k\Delta x - 1)
$$
$$\tag{18}$$

Noting that the coefficients of this quadratic polynomial in $\xi$ are all real (and that the leading coefficient is positive) we may use the Root Amplitude Lemma [3] to show that the stability condition $|\xi| \leq 1$ is obtained if and only if the following two inequalities are satisfied:

$$
\begin{aligned}
A \geq C \qquad\quad & : a\Delta t \geq 2\tfrac{b\Delta t}{\Delta x^2}\sum_{k=1}^{N}(\cos\omega_k\Delta x - 1) \\
A + C \geq |B| \quad & : 2 + 2\tfrac{b\Delta t}{\Delta x^2}\sum_{k=1}^{N}(\cos\omega_k\Delta x - 1)] \geq \left| 2 + 2\tfrac{b\Delta t}{\Delta x^2}\sum_{k=1}^{N}(\cos\omega_k\Delta x - 1)] \right|
\end{aligned} \tag{19}
$$

The upper inequality is satisfied as $a, b, \Delta t, \Delta x \geq 0$. The bottom inequality is satisfied if and only if its left hand side is non-negative, leaving us with

$$
\Delta x^2 \geq b\Delta t\sum_{k=1}^{N}(\cos\omega_k\Delta x - 1) \tag{20}
$$

which, in turn, must be satisfied for all digital frequencies $\omega_k$. This yields the result specified in the Theorem.

$\square$

# C    PDE Acceleration Applied to Weight Tensors in CNNs

We now specify how Accelerated PDE are applied to CNNs. For a given layer, $\theta$ will denote the convolutional filters or batch norm weight tensors. In the former case, $\theta$ is a 4D tensor, i.e., $\theta$ is a function of $h, w$ (the spatial dimensions of the filters) and $i, o$ (the input and output channel directions). In the case of batch norm weights, $\theta$ is a 1D tensor of the scalings or a 1D tensor of the bias. See Figure 2.To implement our scheme as a modification of SGD-Nesterov, we specify the computation of the diffusion term $\nabla^2 v$, as the other terms are the same as SGD-Nesterov. In the case of the convolutional filter weight tensor, we can compute the Laplacian in all or a subset of directions. In experiments, we explore the 2D spatial diffusion, and 1D output and input as well as the 1D diffusions of the batch norm tensors. The Laplacian is approximated in these directions with central differences, using (5), but only in the desired subset of the $N$ directions. A Neumann boundary condition is used.

Our implementation is shown in Algorithm 1, which over standard SGD Nesterov requires just an extra diffusion computation, as well as a few formulas to compute $\beta$ to ensure a stable scheme.

---

[2]One could analyze the effects of the inhomogeneous component within this framework by linearizing the gradient, which may have effects on the step size condition. In practice, we approximate the gradient with a stochastic gradient, which we treat as zero mean noise. Our analysis can be thought of as the analysis where the diffusive component dominates the effect of this noise.

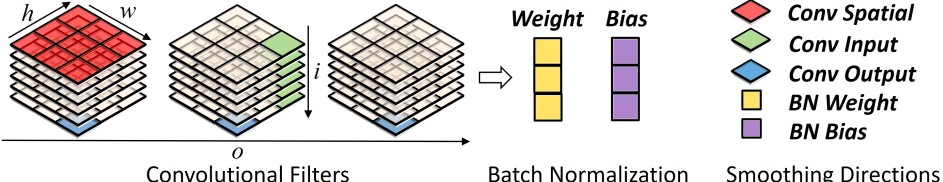

Figure 2: A layer in a CNN visualizing weight tensors (the 4D convolutional filter and the batch normalization weight tensors). Various directions in the tensors are shown. The diffusion term acting on the velocity tensors in our Accelerated PDE can be defined along these directions, and we explore these directional diffusions in the experiments.

---

Algorithm 1: Nesterov SGD with Diffusion (Semi-Implicit Euler Scheme)

---

**Input:** momentum $\gamma$, learning rate $\eta$, max iterations $M$, dimensions of diffusion $N$
**Initialize:** $\theta^0, v^0$
1: Compute $L^0 = \nabla^2 v^0$ using (5)
2: **while** $n < M$ **do**
3:     Compute $\Delta t = \sqrt{\frac{2\eta}{\gamma+1}}$, and $a = \frac{2-2\gamma}{\sqrt{2\eta(\gamma+1)}}$
4:     Compute $b_{\max} = \frac{1}{2N\Delta t}$
5:     Compute $\beta = \frac{2b_{\max}\Delta t}{2+a\Delta t}$
6:     Update $v^{n+1} = \gamma v^n + \nabla_\theta J(\theta^n) + \beta L^n$
7:     Compute $L^{n+1} = \nabla^2 v^{n+1}$ using (5)
8:     Update $\theta^{n+1} = \theta^n - \eta[\nabla J(\theta^n) + \gamma v^{n+1} + \beta L^{n+1}]$
9:     Update $n \leftarrow n + 1$
10: **end while**

---

## D   Image classification experiments

Experiments are shown here on CNNs (DenseNet [9], WideResNet[28]) and datasets (Cifar10, Cifar100, SVHN, Fashion-MNIST) to evaluate the benefit of adding diffusion. We use the cosine annealing schedule with warm restarts [12] for all experiments and Auto Augment [6] is also tested.

The diffusion is simply added to the weight of Batch Normalization layer, which is a 1-D tensor. So the diffusion term in equation (7) is a 1-D spatial Laplacian and we use periodic padding for boundary condition. And for the first epoch, we use the maximum stable diffusion computed from equation 11 and we reduce the diffusion linearly to zero until the last epoch to achieve the best performance. Other parameters still use the standard SGD update scheme.

As seen in Table 4, diffusion outperforms standard SGD with statistical significance in these cases. The experimental results represent the average of 5-10 independent trials. This is just the preliminary result we have on the practical training schedule. However, we have not done enough research on how to fit the diffusion in the practical case. There is more potential for the benefit of diffusion and we will further explore it.

| CNN | Augmentation | SVHN | | Fashion MNIST | |
|---|---|---|---|---|---|
| | | SGD | APDE | SGD | APDE |
| DenseNet-BC-100-12 | Basic | 98.18 ± 0.0030 | **98.25** ± 0.0006 | 95.22 ± 0.0100 | **95.38** ± 0.0051 |
| | AA | 98.45 ± 0.0013 | **98.51** ± 0.0019 | 95.81 ± 0.0171 | **95.87** ± 0.0053 |
| WideResNet-16-4 | Basic | 98.34 ± 0.0007 | **98.37** ± 0.0007 | 95.59 ± 0.0008 | **95.74** ± 0.0070 |
| | AA | 98.72 ± 0.0003 | **98.76** ± 0.0008 | 95.87 ± 0.0074 | **95.90** ± 0.0053 |

Table 3: Test accuracy for SVHN and Fashion MNIST experiment

| CNN | Augmentation | Cifar10 | | Cifar100 | |
|---|---|---|---|---|---|
| | | SGD | APDE | SGD | APDE |
| ResNet56 | Basic | 94.42 ± 0.0253 | **94.57** ± 0.0081 | 72.83 ± 0.1344 | **73.00** ± 0.1600 |
| | AA | 95.27 ± 0.0188 | **95.53** ± 0.0091 | 76.24 ± 0.0642 | **76.28** ± 0.0151 |
| WideResNet-28-10 | Basic | 96.25 ± 0.0162 | **96.49** ± 0.0100 | 81.26 ± 0.0264 | **81.51** ± 0.0377 |
| | AA | 97.35 ± 0.0050 | **97.52** ± 0.0034 | 83.70 ± 0.0193 | **83.80** ± 0.0263 |

Table 4: Test accuracy for Cifar experiment

# E   PYTORCH NESTEROV SGD ALGORITHM

In the official document, the authors claim they implement a modified version of Nesterov, which has the following form:

$$\begin{cases} v^{n+1} & = \gamma v^n + \nabla_\theta J(\theta^n - \gamma\eta v^n) \\ \theta^{n+1} & = \theta^n - \eta v^{n+1} \end{cases} \tag{21}$$

Define $\theta'^n = \theta^n - \gamma\eta v^n$
Then

$$\begin{cases} v^{n+1} & = \gamma v^n + \nabla_{\theta'} J(\theta'^n) \\ \theta'^{n+1} & = \theta^{n+1} - \gamma\eta v^{n+1} = \theta^n - \eta v^{n+1} - \gamma\eta v^{n+1} = \theta'^n + \gamma\eta v^n - \eta v^{n+1} - \gamma\eta v^{n+1} \end{cases} \tag{22}$$

And $\gamma\eta v^n - \eta v^{n+1} = -\eta(v^{n+1} - \gamma v^n) = -\eta\nabla_{\theta'} J(\theta'^n)$
So the original form could be written as

$$\begin{cases} v^{n+1} & = \gamma v^n + \nabla_{\theta'} J(\theta'^n) \\ \theta'^{n+1} & = \theta'^n - \eta(\nabla_{\theta'} J(\theta'^n) + \gamma v^{n+1}) \end{cases} \tag{23}$$

which is the algorithm implemented in the PyTorch SGD optimizer.

