# OpenReview forum: "Accelerated PDEs for Construction and Theoretical Analysis of an SGD Extension"
_NeurIPS.cc/2021/Workshop/DLDE — DLDE Workshop -- NeurIPS 2021 Poster_

### Official Review · Reviewer_gKy3 · 2021-10-01
**Great work! The paper contains the preliminary results for a novel framework that is proposed to address optimization with PDEs**

**Confidence:** 1

**Review:**

This is a very interesting work that enabled me to shed light on the applicability of neural differential equations to the optimization of neural networks. Taking into account my limited knowledge of this specific field, the paper seems to be novel and original providing contributions with preliminary results to the field.

Comments:

Construction of the two function spaces $U$ and $V$, for the weights and velocity, is in accordance given the relation to the SGD algorithm.

Section 2 is appreciated for an understanding of the rest of the paper.

I did not understand/found the validation in the paper of the 4th contribution listed at the end of the introductory section.

Overall the methods section of the paper is very well described, as well as the background of section 2. However, I felt that the last section could be improved to better clarify the reader. Nonetheless, Figure 1 is appreciated and provides a stability analysis of the method.

Please double check if the definition of $w=\theta$ in equations 8 and 9 is needed, it appeared to be redundant. If so, replace $\theta$ with $w$ in those equations.

The accuracy performance presented in the appendix, should be presented in the main sections of the paper, since people often go directly to the results because it provides readers that are not from the optimization field to get a grip of the impact of this work.

Improvements:

Would be nice to have a comparison of the training time of both SGD and your framework, for the sake of completeness.

Providing the accuracy along with the deviation in Table 3 would be appreciated. Further, the statistical significance superiority of the proposed method against SGD  claim should be provided along with the respective p-values, which can greatly describe the impact of your work. Typically, one does not provide the p values if the results are not statistically significant.

Corrections:

"hyper-parameters" -> "hyperparameters"

"... in all $N$ direction)." -> "... in all $N$ directions)."

Suggestions:

- in theorem 1: might be more concise to start a new paragraph outside of the theorem environment at "The mapping of SGD-Nesterov..."

**Score:**

4: Very good paper

---

### Official Review · Reviewer_Cg4i · 2021-10-03
**Interesting theoretical framework to understand variants of SGD with momentum.**

**Confidence:** 4

**Review:**

### Summary

The paper proposes a theoretical framework to understand Nesterov momentum in SGD, that can potentially predict its behavior.

### Comments
**pros**: The theoretical framework is clean and well explained.

**cons**:

L107: Doesn't Nesterov imply evaluating the gradient in the momentum on the future expected position? As far as I understand that would make your work an extension of SGD with momentum, rather than an extension of SGD with Nesterov momentum.

L131: Does it mean that the rest of the network is trained on plain SGD, while BN is trained on your SGD variant?

L145: It would be nice to see the std of the results. I assume you report accuracies, but it's not stated, and that leads to confusion.

Table 1: Best accuracies (if they are accuracies) seem to be for 4*\beta_max, which seems to contradict the theory.

**Score:**

4: Very good paper

---

### Official Review · Reviewer_Jb5u · 2021-10-11
**Interesting extension to SGD motivated from discretized PDEs**

**Confidence:** 2

**Review:**

The paper discusses relationships between SGD-momentum and discretized PDEs to suggest a new version of SGD with added diffusion term. The authors provide a stability criterion for the strength of diffusion and verify it numerically. Results on performance improvements on benchmark problems are also included.

I think that the main idea and theoretical results are a strong contribution and should be presented at the workshop. For the numerical results, some numbers on the added computational cost of smoothing would be useful. For table 3, some measure of variance would be good. More generally, I would be interested not only in the reported final performance of the trained networks, but even more so in learning dynamics, i.e. does the suggested SGD variant converge faster, does it show improvements rather in earlier or later training stages?

The paper overall is written clearly. The abstract could be easier to read if the authors would define some abbreviation (SGD, CNNs) only in the main text. The introduction extensively lists related work and does a good job situating the current study within the current field, but feels a bit hard to read without following up all of those citations – one or two more sentences summarizing the cited papers could help to guide unfamiliar readers?

Minor

Line 81: should be “V” instead of “v” ?

Equations 6,7: for Nesterov don’t you want J(theta*), where theta* is the look-ahead point?

Table 1&2: are these test accuracies, and if so, could you state that as you did for table 3?

Figure 2: nice figure overall, but does the empty white arrow between the panels for convolutional filters and batch normalization serve any purpose? I found it rather confusing.

Line 264: please specify “statistical significance”

**Score:**

3: Good paper

---

### Decision · Program_Chairs · 2021-10-15

**Decision:**

Accept (Poster)

**Comment:**

Reviews were positive, and reviewers were eager to hear more about the work. The reviewers also had many questions and comments, which the authors may wish to address to maximize the accessibility and impact of their poster.